# Photodynamic Antibiofilm and Antibacterial Activity of a New Gel with 5-Aminolevulinic Acid on Infected Titanium Surfaces

**DOI:** 10.3390/biomedicines10030572

**Published:** 2022-02-28

**Authors:** Morena Petrini, Silvia Di Lodovico, Giovanna Iezzi, Luigina Cellini, Domenico Tripodi, Adriano Piattelli, Simonetta D’Ercole

**Affiliations:** 1Department of Medical, Oral and Biotechnological Science, University G. D’Annunzio of Chieti, Via dei Vestini 31, 66013 Chieti, Italy; gio.iezzi@unich.it (G.I.); tripodi@unich.it (D.T.); apiattelli51@gmail.com (A.P.); simonetta.dercole@unich.it (S.D.); 2Department of Pharmacy, University of Chieti, Via dei Vestini 31, 66013 Chieti, Italy; silvia.dilodovico@unich.it (S.D.L.); l.cellini@unich.it (L.C.); 3School of Dentistry, Saint Camillus International University of Health and Medical Sciences, Via di Sant’Alessandro 8, 00131 Rome, Italy; 4Faculty of Dental Medicine, University of Belgrade, 11000 Belgrade, Serbia; 5Fondazione Villa Serena per la Ricerca, 65013 Città Sant’Angelo, Italy; 6Casa di Cura Villa Serena del Dott. L. Petruzzi, 65013 Città Sant’Angelo, Italy

**Keywords:** photodynamic therapy, aminolevulinic acid, titanium, peri-implantitis

## Abstract

The use of a new gel containing aminolevulinic acid and red light (ALAD–PDI) was tested in order to counteract bacterial biofilm growth on different titanium implant surfaces. The varying antibacterial efficacy of ALAD–PDI against biofilm growth on several titanium surfaces was also evaluated. A total of 60 titanium discs (30 machined and 30 double-acid etched, DAE) were pre-incubated with saliva and then incubated for 24 h with *Streptococcus oralis* to form bacterial biofilm. Four different groups were distinguished: two exposed groups (MACHINED and DAE discs), covered with *S. oralis* biofilm and subjected to ALAD + PDI, and two unexposed groups, with the same surfaces and bacteria, but without the ALAD + PDI (positive controls). Negative controls were non-inoculated discs alone and combined with the gel (ALAD) without the broth cultures. After a further 24 h of anaerobic incubation, all groups were evaluated for colony-forming units (CFUs) and biofilm biomass, imaged via scanning electron microscope, and tested for cell viability via LIVE/DEAD analysis. CFUs and biofilm biomass had significantly higher presence on unexposed samples. ALAD–PDI significantly decreased the number of bacterial CFUs on both exposed surfaces, but without any statistically significant differences among them. Live/dead staining showed the presence of 100% red dead cells on both exposed samples, unlike in unexposed groups. Treatment with ALAD + red light is an effective protocol to counteract the *S. oralis* biofilm deposited on titanium surfaces with different tomography.

## 1. Introduction

The role of bacterial biofilm in the etiology and the development of peri-implant disease has been demonstrated [1]. Indeed, patients who do not practice proper plaque control are 3.8 times more affected by peri-implantitis compared to those with good oral hygiene habits [2]. Peri-implantitis is a heterogeneous and complex infection. The microbial ecosystem is composed of Gram-negative periodontal pathogens, such as *Porphyromonas gingivalis* and *Prevotella intermedius/nigrescens*, and bacterial species that are not associated with periodontitis [3,4].

Peri-implant disease is distinguished between the reversible form, denoted mucositis, and the irreversible one, denoted peri-implantitis [5]. A recent Consensus report highlighted that the only risk factors of peri-implantitis for which exist robust scientific evidence are poor plaque control, history of severe periodontitis, and lack of regular care [6].

The role of titanium surfaces in the modulation of bacterial interaction has been extensively investigated. It is known that rough surfaces, when exposed to the oral cavity, increase the risk of plaque accumulation [7]. However, moderately rough surfaces showed a lower prevalence of peri-implantitis compared to minimally or maximally rough ones [8]. Recently, nanoscale investigations showed that nanopatterning plays a fundamental role in modulating bacterial colonization, and very rough surfaces, such as double-acid-etched (DAE) surfaces, are associated with a lower bacterial count at 24 and 48 h of incubation [9].

Currently, research is focused on the development of novel surfaces that could counteract biofilm accumulation in order to prevent the onset of peri-implantitis, and on novel treatment strategies to decontaminate titanium and its alloys without damaging the surface tomography [9,10,11].

In our recent studies, we focused on photoinactivation using light-emitting diodes (LEDs) against common pathogens of the oral cavity. In particular, near-infrared light (NIR) showed an antibacterial effect against *Enterococcus faecalis* and *Pseudomonas aeruginosa* [12,13,14,15]. This was confirmed over time and was effective against both planktonic and sessile bacteria [12,13,14,15]. Moreover, low-intensity red and NIR lights are characterized by an anti-inflammatory action that is particularly useful in oral surgery, as shown by many in vivo and in vitro studies. These therapeutic effects are dependent on the presence of specific targets in the tissues, the endogenous photosensitizers, which, irradiated by specific wavelengths, are able to induce the cellular response [16,17,18].

The presence of these molecules depends on the bacterial species, growth conditions, bacterial strains and other factors; therefore, the photoinactivation is not constant [19]. By comparison, the addition of an exogenous photosensitizer, such as the newly formulated gel Aladent (Alpha Strumenti, Melzo, Milan Italy), ALAD containing 5-aminolevulinic acid (ala), followed by red light irradiation showed interesting results against both Gram-positive and Gram-negative bacteria, including the periodontal pathogen *P. gingivalis* and yeast *Candida albicans* [20,21,22].

*Streptococcus oralis*, a Gram-positive bacterium, is one of the early colonizers that, upon interacting with the acquired pellicle deposited on biomaterials, provides the basis for polymicrobial biofilm formation due to subsequent colonization by facultative and obligate anaerobic microorganisms [23]. This shift in the composition of the microbial ecosystem can lead to a local host inflammatory response in the peri-implant tissues, and depending on the presence of other risk factors, this could lead to reversible peri-mucositis or could cause peri-implantitis characterized by irreversible bone loss [6,24,25].

*Streptococcus oralis* is widely used for in vitro research on titanium surfaces. In this study, we present the effect of photodynamic therapy involving the use of a novel gel containing aminolevulinic acid combined with red LED irradiation (ALAD–PDI) on *S. oralis* biofilm, grown on MACHINED and DAE surfaces. This bacterium was chosen as a possible example of an oral microorganism; further studies with a more relevant mixture of bacteria would have to be conducted for potential clinical applications.

The aim of this preliminary study was to evaluate the effects of ALAD–PDI on *Streptococcus oralis* biofilm growth on two different titanium surfaces, MACHINED and DAE. The secondary outcome was to evaluate whether the tested protocol was equally effective on both titanium surfaces.

## 2. Materials and Methods

The protocol for this in vitro study was implemented in accordance with the Enhancing the QUAlity and Transparency Of health Research (EQUATOR) guidelines, in particular the Standards for Reporting Qualitative Research SRQR [26].

A total of 60 titanium discs, grade IV (ASTM F67) (Resista, Omegna (VB), Italy), 5 mm (diameter) × 2 mm (thickness), were used in this study.

The discs were manufactured using the same material, but were characterized by two different surfaces, as shown in Figure 1A,B:MACHINED DISCS: titanium turned surfaces, obtained from the milling of a bar;DAE DISCS: double-acid-etched (DAE) surfaces produced using a double mixture of nitric, hydrochloric, and hydrofluoric acid, and final neutralizing buffer.

After manufacturing, all discs were subjected to ultrasonic cavitation in a basic solution, and a cold argon plasma reactor decontamination. Then, before the experiment, all discs were immersed for 60 min in 75% ethanol, and left to dry in a test tube previously sterilized via UV irradiation for 30 min.

### 2.1. Saliva Sampling

The spitting method was used to sample saliva from four healthy donors, as previously described [27,28,29]. The use of human saliva for this in vitro study was approved by the local Ethics Committee.

The processing of sampled saliva was performed in accordance with previous literature. Briefly, in order to remove debris and ensure sterility, it was subjected to cycles of centrifugation and filtration [30].

Then, the discs were put on 96 multi-well polystyrene microtiter plates and immersed for 2 h in saliva [9,11,31].

### 2.2. Microbial Strain and Biofilm Development

A clinical strain of *Streptococcus oralis* CH 05, isolated from a saliva sample from a healthy individual and collected at the Department of Medical, Oral and Biotechnological Science, University G. D’annunzio of Chieti; Via dei Vestini 31, 66013, Chieti, Italy, was used for the study, as previously described [32].

To evaluate the effect of the tested discs on *S. oralis* CH 05 growth, 200 µL of standardized broth culture (OD_600_ = 0.12, corresponding to 9 × 10^6^ CFU/mL) was dispensed on saliva-coated discs and incubated for 24 h + 24 h at 37 °C in an anaerobic chamber (80/10/10, N_2_/H_2_/CO_2_; Don Whitley Scientific Ltd., Shipley, UK; International PBI SpA) [22,25].

In detail, four different groups were treated as follows:MACHINED: machined discs preincubated in saliva for 2 h and inoculated with *S. oralis* to permit biofilm formation, 24 h + 24 h (machined positive controls);MACHINED + ALAD: machined discs preincubated in saliva for 2 h, inoculated with *S. oralis* (biofilm formation for 24 h) and then exposed to ALADENT gel and red LED irradiation (ALAD–PDI), and incubated for a further 24 h before microbiological analysis, as shown in Figure 1C–E;DAE: DAE discs preincubated in saliva for 2 h and incubated with *S. oralis* to permit biofilm formation, 24 h + 24 h (machined positive controls);DAE + ALAD: DAE discs, preincubated in saliva for 2 h, inoculated with *S. oralis* (biofilm formation for 24 h) and then subjected to ALAD–PDI, and incubated for a further 24 h before microbiological analysis.Non-inoculated titanium discs used as negative controls.

### 2.3. ALADENT Gel (ALAD) and Irradiation Parameters: ALAD–PDI

The ALADENT gel (ALAD) (Alpha Strumenti, Melzo, Milan, Italy), used in this experiment contains 5% of 5-delta aminolevulinic acid and other components, covered by a patent [20]. This substance is liquid at a temperature less than 28 °C and a gel at higher temperatures. Experimentation was performed at room temperature, under a microbiological incubator, and 200 µL of ALAD was pipetted onto each titanium disc of the exposed groups, after 24 h of bacterial inoculation, and left to incubate in darkness for 45 min.

Then, an AlGaAs power LED device (TL-01), characterized by a red light with a wavelength of 630 nm ± 10 nm FHWM nm (Alpha Strumenti, Melzo, Milan, Italy), was used as irradiation source for 7 min at 380 mW/cm^2^. The tip of the red LED was maintained perpendicular to the upper surfaces of the discs at a constant distance of 0.5 mm using a specific apparatus, as previously described and shown in Figure 1C–E [20].

### 2.4. Microbiological Analysis

After a further 24 h of incubation in an anaerobic chamber, microbiological analyses were performed in all tested groups in order to determine:Colony Forming Units (CFUs) enumerationBiofilm biomass quantificationCell viability via live/dead analysis.Scanning electron microscope observation

Negative controls were non-inoculated discs alone and combined with gel (ALAD), without the broth cultures. All negative controls were incubated for 24 h in an anaerobic environment, after which one was treated for 45 min with ALAD and the other left in an aerobic environment, and then all discs were incubated for another 24 h in an anaerobic environment.

Positive controls were incubated for 24 h in an anaerobic environment and then treated for 45 min with ALAD, irradiated for 7 min with red light, and left to incubate for another 24 h in an anaerobic environment.

#### 2.4.1. Determination of Colony-Forming Units (CFUs)

For the enumeration of adherent cells, in all conditions, the planktonic phase was removed and each disc was transferred into a tube with 1 mL of PBS, sonicated in a 4- kHz ultrasonic bath (Euronda, Sandrigo, Italy) for 4 min and then vortexed for 2 min. The obtained suspension was diluted and spread on Tryptic Soy Agar (TSA). The plates were incubated at 37 °C in an anaerobic chamber. Before the spreading on TSA, microscopic observations with live/dead staining confirmed that the diluted cells were disaggregated and viable (data not shown).

#### 2.4.2. Biofilm Biomass Assay

For biofilm biomass quantification, after 24 h + 24 h of incubation in an anaerobic chamber, exposed and unexposed discs were treated based on our previous studies [9,20]. The discs were washed from dead cells, air-dried, and stained with 0.1% Crystal Violet. Then, the discs were resuspended in 200 μL ethanol. After the removal of the disc, elution was measured at OD_570_ using an ELISA reader (SAFAS, Munich, Germany).

#### 2.4.3. Viability Test

The cell viability on each tested disc was examined using a BacLight LIVE/DEAD Viability Kit (Molecular Probes, Invitrogen detection technologies, Eugene, OR, USA) as instructed by the manufacturer and visualized under a fluorescent Leica 4000 DM microscope [11,20]. The control group, unexposed to ALAD–PDI, was used to evaluate the effect of each variable: the type of titanium surface and the effect of the photodynamic therapy. The enumeration was performed by three blinded microbiologists (S.D.L., S.D., and L.C.) using image analysis software LEICA QWin 3 (Leica Microsystems Inc., Wetzlar, Germany) through the examination of at least 10 random fields of view each [11]. The percentages of viable and dead cells were calculated.

### 2.5. Scanning Electron Microscope Observations (SEM)

Before proceeding with SEM observations, all discs were fixed with glutaraldehyde, dehydrated with ascending ethanol alcohol concentrations, and gold-sputtered, as previously described [9,11].

A Phenom ProX scanning electron microscope (Phenom-World B.V., Eindhoven, The Netherlands) was used to characterize the samples at microscale, at 3900× and 6100× magnifications, using the following parameters FOV: 559 µm, Mode: 15 kV—Map, Detector: BSD Full.

Adobe Photoshop (Park Avenue, San Jose, CA, USA) version 9.0 was used to color in red the bacterial cells (opacity 32%), as shown in Figure 3.

### 2.6. Statistical Analysis

The software SPSS for Windows version 21 (IBM SPSS Inc., Chicago, IL, USA) was used for statistical analysis of the results. One-way ANOVA testing was performed to verify the presence of statistically significant differences, and in the case of *p* < 0.050 the post-hoc Fisher’s least significant difference (LSD) test was performed. The LSD is a two-step testing procedure for pairwise comparisons of several treatment groups [33]. *p*-values < 0.050 were considered significant.

## 3. Results and Discussion

DAE surfaces showed significantly lower numbers of *S. oralis* CFUs compared to MACHINED, in accordance with our previous studies. [9,11].

Microbiological evaluation displayed significantly less bacterial growth on unexposed DAE surfaces with 7.255 ± 0.165 log_10_ CFU/mL compared to 7.423 ± 0.072 log_10_ CFU/mL on the MACHINED ones. As shown in Figure 2A,B, both surfaces exposed to ALAD–PDI demonstrated a significant decrease in bacterial load, with statistically significant differences, *p* < 0.050. In particular, the number of cells on MACHINED + ALAD surfaces was 6.427 ± 0.174 log_10_ CFU/mL, an 89% bacterial load reduction compared to the unexposed MACHINED ones. DAE + ALAD surfaces showed 6.618 ± 0.130 log_10_ CFU/mL, a 77%bacterial load reduction compared to the unexposed DAE.

The biofilm biomass, Figure 2C,D, displayed significantly higher values on unexposed DAE surfaces, 0.858 ± 0.100, compared to MACHINED, 0.6431 ± 0.125. These values significantly decreased, *p* < 0.050, in the exposed discs, to 0.255 ± 0.097 and 0.447 ± 0.101 in MACHINED + ALAD and DAE + ALAD, respectively.

The different bacterial interactions with the titanium surfaces confirmed the necessity of testing specific methods of decontamination for both surfaces to verify whether ALAD–PDI was effective in both conditions. Moreover, treatment with ALAD–PDI significantly reduced the *S. oralis* biomasses produced on both tested surfaces. The CFU/mL value and the biomass value were in disagreement; however, they are two different methods to evaluate the effect of *S. oralis* growth on surfaces alone or combined with ALAD–PDI. CFU count is a direct method for determining the number of viable cells in a biofilm; biomass quantification is an indirect determination of biofilm growth that includes the matrix, DNA, RNA, polysaccharides, or metabolites attached to the cells of the microbial biofilm [34].

The scanning electron microscope observations showed the presence of relevant bacterial biofilm on MACHINED surfaces, independently of exposure or unexposure to ALAD–PDI, Figure 3. DAE surfaces showed less bacterial presence on both unexposed and exposed surfaces.

In exposed samples, aggregated cells were detected, and non-adherent bacteria on the titanium surface were probably trapped in the gel matrix.

The ALAD gel acted as a glue that trapped bacteria before inactivating them via photodynamic therapy (ALAD–PDI). This phenomenon allowed for great decontamination of the titanium surfaces, but implied that after using ALAD for the treatment of peri-implantitis, it would be desirable to add a mechanical device for the removal of the dead bacteria from the implant surfaces. In addition, ALAD–PDI exerted a significant bactericidal effect compared to the unexposed samples, as demonstrated in Figure 4. Images obtained by fluorescent microscopy demonstrated the presence of a higher bacterial load on exposed and unexposed MACHINED discs relative to DAE ones. The live/dead staining showed that 85% and 75% of cells were green (live) on unexposed MACHINED and DAE discs, respectively. By comparison, 100% of detected cells were red (dead) on both MACHINED + ALAD and DAE + ALAD discs (Figure 4, histogram).

These results are in accordance with our previous studies, in which we tested the effect of ALAD–PDI on both Gram-positive and negative bacteria [20,35]. By comparison, previous literature that used different formulations of aminolevulinic acid followed by photodynamic therapy (ala–PDI) showed contradictory results, especially on Gram-negative bacteria [36,37]. It was shown that the increase in reactive oxygen species (ROS) induced by ala–PDI promotes intracellular biopolymer leakage, photocleavage on genomic DNA, cytoplasm denaturing, and envelope injury [38].

The production of ROS is mediated by red LED irradiation of porphyrin IX (ppi IX). This photosensitizer is produced during gel incubation intracellularly, started by its precursor, the ala. Mammalian cells are able to transform ppi by means of enzymes such as ferrochelatase, and promote the formation of the final product, the heme-group, so that after a period of incubation, the presence of ppi on these cells is reduced [39,40,41,42]. Cancer cells and bacteria, characterized by higher turnover or by the lack of these specific enzymes, accumulate a greater amount of ppi, and light irradiation at specific wavelengths promotes the production of ROS and consequently efficient and specific cytotoxic activity [43,44]. The amount of porphyrin (ppi) formation is not directly proportional to the ala concentration: Bohm et al. found a peak in ppi formation and a reduction in CFUs with 1 mM ala in both *Staphylococcus aureus* and *Staphylococcus epidermidis*, but a further increase of ala concentration was associated with decreases in ppi and in the antibacterial activity provided by ala–PDI [45].

The encouraging results, both antibacterial and antifungal, achieved using ALAD–PDI relative to ala–PDI, could be the consequence of three factors that act concomitantly: the production of reactive oxygen species (ROS), the pH value of 3.5, and the presence of preservatives, such as potassium sorbate and sodium benzoate [20,21,46]. Greco et al. showed that the antifungal effect obtained by ALAD–PDI was highly influenced by the ALAD’s pH [21].

The antibacterial effects of ALAD gel followed by red LED irradiation are well known, but, in this study, this novel gel for ALAD–PDI was tested for the first time on bacterial biofilm growth on titanium surfaces, rather than in planktonic bacteria [47,48]. As previously described, the efficacy of other treatments containing aminolevulinic acid and light irradiation (ala–PDI) on counteracting bacterial biofilm is very variable. Indeed, ppi formation differs among bacterial species, and depends on the particular growth conditions. Similar ppi production can be difficult to detect because some bacteria species produce non-fluorescent porphyrins that can be equally activated by light irradiation [45]. Bohom et al. showed that ala–PDI had effective antibiofilm action against Gram-positive biofilms; however, Gram-negative biofilms were quite refractory and showed no ala incorporation [45].

Recently, Liu et al. showed an inhibitory effect of ala–PDI with another compound containing aminolevulinic acid on *Propionibacterium acnes* biofilm [47], but their results were significant only when using high light doses, starting from 100 J/cm^2^. Li et al. also showed a drastic reduction in both methicillin-resistant *S. aureus* and methicillin-resistant *S. epidermidis* survival within biofilms, and disruption of biofilms using ala–PDI and laser irradiation and power intensity higher starting from 100 J/cm^2^ [42]. The effectiveness of ala–PDI treatment against *Candida albicans* biofilm was recently confirmed by Shi et al.; however, the ala-based compound used in this study was incubated for 5 h before proceeding with red laser irradiation at a light intensity of 300 J/cm^2^ and fluence rate of 100 mW/cm^2^ for 50 min [48]. By comparison, the ALAD–PDI demonstrated effective significant inhibition of *S. oralis* biofilm (*p*-value < 0.050) with a shorter incubation period and lower power intensity, at 380 mW/cm^2^.

The clinical implications of these results are very important, because the presented protocol could be very useful for the decontamination of the titanium surfaces of dental implants affected by peri-implant disease without risk of overheating the fixtures or burning the surrounding tissues. Many methods and treatment protocols for the decontamination of implants and biofilm eradication are described in the literature: mechanical devices for plaque removal, such as titanium brushes and Teflon tips; chemical compounds, such as acids or antibacterial compounds; and physical devices, such as lasers [4]. Current literature fails to show which method is most effective; however, it suggests that the chances of successfully treating peri-implant disease decrease if the diagnosis is delayed [4].

The ALAD–PDI protocol is a further aid for bactericide without the necessity of applying antibiotics, the use of which is questionable and, in addition, contributes to antibiotic resistance, a worldwide health problem of significant importance. The limitation of this study is the use of a mono-species bacteria model. Moreover, although *S. oralis* is commensal in the oral cavity, recent literature showed that its biofilm is able to severely damage fibroblasts (HGFs) grown on titanium discs by stimulating a stress response and production of inflammatory mediators [49].

We believe that the encouraging results obtained by this preliminary study will permit the carrying out of more complex protocols involving multi-species trials.

## 4. Conclusions

ALAD–PDI promoted significant bacterial biofilm reduction and titanium decontamination, both on MACHINED and DAE surfaces. If these results are confirmed by further studies, this protocol could be proposed as a novel method of implant disinfection for fixtures affected by peri-implant disease. Considering the antibiofilm activity shown by ALAD–PDI, another possible application of this protocol could be the treatment of chronic wounds.

## Figures and Tables

**Figure 1 biomedicines-10-00572-f001:**
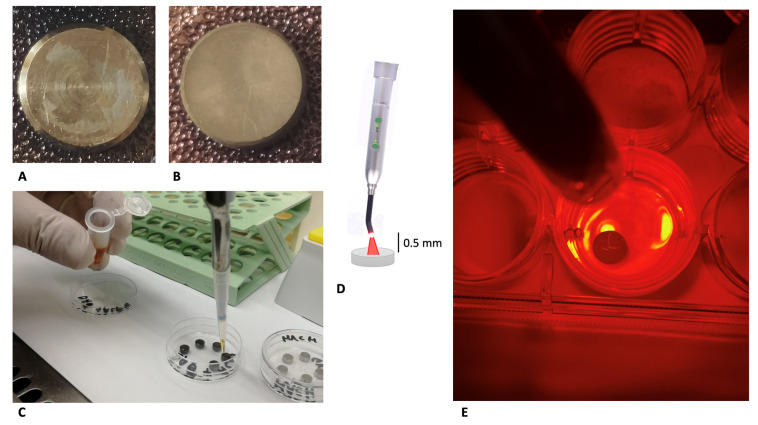
Study design: machined (**A**) and DAE (**B**) discs before experimentation. Application of ALAD (**C**) to the discs following pretreatment with human saliva and incubation for 24 h with *Streptococcus oralis*. Irradiation of the discs for 7 min with a 630 nm LED, illustration (**D**), and photograph (**E**).

**Figure 2 biomedicines-10-00572-f002:**
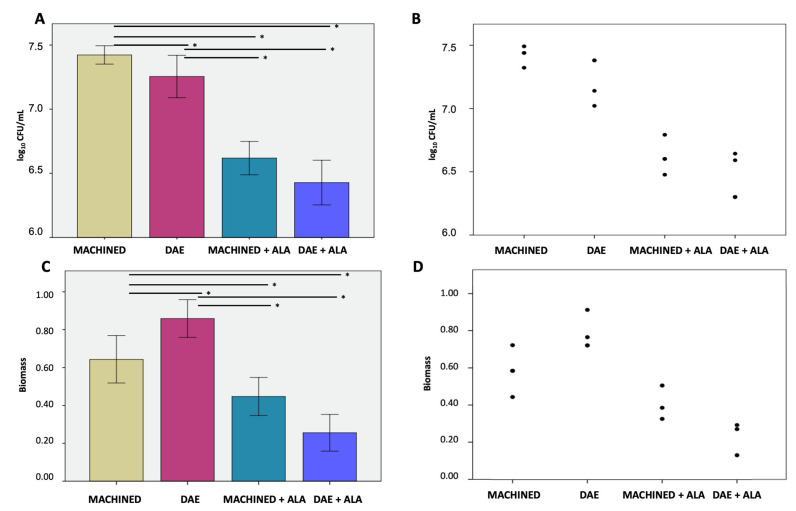
Colony-forming units (**A**) and relative dot plot (**B**); biofilm biomass (**C**) and relative dot plot (**D**) of *Streptococcus oralis* biofilm grown for 24 h + 24 h on MACHINED and DAE surfaces with or without being exposed to ALAD–PDI. Negative controls showed no CFUs and no biofilm biomass (data not shown). * *p*-value < 0.050.

**Figure 3 biomedicines-10-00572-f003:**
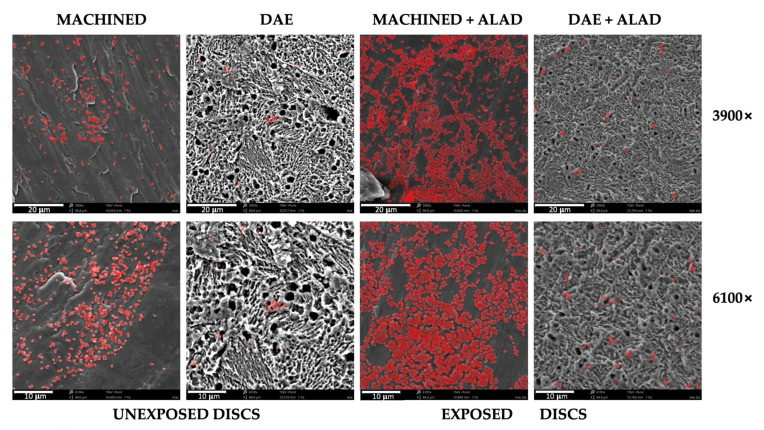
Scanning electron microscope observations at different magnifications, 3900× and 6100×, of *Streptococcus oralis* biofilm grown for 24 h + 24 h on MACHINED and DAE discs unexposed and exposed to ALAD gel and red LED. Bacteria are colored in red; lighter colors correspond to higher bacterial density.

**Figure 4 biomedicines-10-00572-f004:**
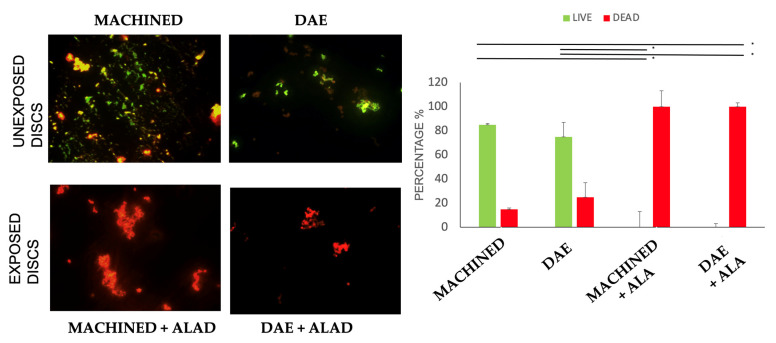
Live/dead staining of *Streptococcus oralis* biofilm grown for 24 h + 24 h on MACHINED and DAE discs unexposed and exposed to ALAD–PDI. Histograms show percentages of viable and dead cells for each exposed group vs. the unexposed samples, obtained with identical methods in every respect except for exposure to ALAD–PDI (* *p*-value < 0.050). Negative controls showed no cells (data not shown).

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
