# Peer review of "Photodynamic Antibiofilm and Antibacterial Activity of a New Gel with 5-Aminolevulinic Acid on Infected Titanium Surfaces"

_biomedicines, 2022, doi:10.3390/biomedicines10030572_

Round 1
Reviewer 1 Report
I understood that this is a revised version for a manuscript that I have not reviewed before.
I think the idea of the research is clear: effect of aminolevulinic acid and red light (ALAD- 16PDI) on the antimicrobial activity of titanium surfaces anf that they significantly decreased the colony forming units.
I have only found some textual errors and I would recommend that the authors revise carefully again the whole text for this kind of mistakes.
Author Response
I understood that this is a revised version for a manuscript that I have not reviewed before. I think the idea of the research is clear: effect of aminolevulinic acid and red light (ALAD- 16PDI) on the antimicrobial activity of titanium surfaces anf that they significantly decreased the colony forming units. I have only found some textual errors and I would recommend that the authors revise carefully again the whole text for this kind of mistakes.- AUTHOR’S ANSWER: Thank you very much for your comment. The manuscript have been checked and corrected.
Reviewer 2 Report
The manuscript describes a novel antibacterial gel that could be used in medicine to protect implants from bacterial biofilms.
The design of the study seems fine. The only issue that may be brought to light here is a relatively small number of repetitions. However, the authors included all necessary information about replicates in the manuscript (e.g. Figure 1). The methodology is standard for such studies and enough to follow the aims set by the authors.
The manuscript is generally well written and, judging by the yellow fields, it was already peer-reviewed in the previous round in which I did not take part. There are some issues that should be attended before the manuscript is recommended for publication.
The introduction lacks information on the used S. oralis strain. There are passages about other microbes, but the importance of this one is missing.
The sentence in lines L68-69 is not clear. What do you mean by "the presence of endogenous photosensitizer depends on the bacterial type...?"
The first paragraph of the results section could be inserted into the introduction.
The conclusion section should be further developed and not only repeat the result. How it can be further used? Why it is important to apply the proposed solution? What are further stages if this idea goes to practice?
I would like to point out also some minor issues
L40 "gram-negative" should be in capital letter
L65 "in vivo" and "in vitro" should be written in italics
L233 "a test tube" - different font
Author Response
The manuscript describes a novel antibacterial gel that could be used in medicine to protect implants from bacterial biofilms. The design of the study seems fine. The only issue that may be brought to light here is a relatively small number of repetitions. However, the authors included all necessary information about replicates in the manuscript (e.g. Figure 1). The methodology is standard for such studies and enough to follow the aims set by the authors.
- AUTHOR’S ANSWER: thank you very much for your comment; as you have said we have used a standard methodology that we have already used in our previous publications.
• D’Ercole, S.; Mangano, C.; Cellini, L.; Di Lodovico, S.; Atalayin Ozkaya, C.; Iezzi, G.; Piattelli, A.;Petrini, M. A Novel 3D Titanium Surface Produced by Selective Laser Sintering to Counteract Streptococcus oralis Biofilm Formation. Appl. Sci. 2021, 11, 11915.
• D’Ercole, S.; Di Lodovico, S.; Iezzi, G.; Pierfelice, T.V.; D’Amico, E.; Cipollina, A.; Piattelli, A.; Cellini, L.; Petrini, M. Complex Electromagnetic Fields Reduce Candida albicans Planktonic Growth and Its Adhesion to Titanium Surfaces. Biomedicines 2021, 9, 1261. https://doi.org/10.3390/biomedicines9091261
• Petrini M, Di Lodovico S, Iezzi G, Cipollina A, Piattelli A, Cellini L, D’Ercole S. Effects of Complex Electromagnetic Fields on Candida albicans Adhesion and Proliferation on Polyacrylic Resin. Appl. Sci. 2021, 11(15), 6786; https://doi.org/10.3390/app11156786
• Petrini M, Giuliani A, Di Campli E, Di Lodovico S, Iezzi G, Piattelli A, D’Ercole S. The Bacterial Anti-Adhesive Activity of Double-Etched Titanium (DAE) as a Dental Implant Surface. Int. J. Mol. Sci. 2020, 21, 8315; doi:10.3390/ijms21218315
-The manuscript is generally well written and, judging by the yellow fields, it was already peer-reviewed in the previous round in which I did not take part. There are some issues that should be attended before the manuscript is recommended for publication. The sentence in lines L68-69 is not clear. What do you mean by "the presence of endogenous photosensitizer depends on the bacterial type...?"
- AUTHOR’S ANSWER: thank you very much for your comment: endogenous photosensitizers are molecules that can be present inside the bacteria, and if these molecules are irradiated by a specific light, are able to start a sequence of events, like the production of Reactive oxygen species, free radicals, or second messengers cascade. However, the presence of these molecules inside the bacterial cells is not constant, but is dependent on some factors, like the bacterial type, growth conditions, bacterial strains, and other factors. The sentence in the text has been changed in: “These therapeutic effects are dependent on the presence in the tissues of specific targets, the endogenous photosensitizers, that, irradiated by specific wavelengths, are able to induce the cellular response [16–18].In particular, the presence of these molecules depends on the bacterial type, growth conditions, bacterial strains, and other factors, so the photoinactivation was not constant”.
The introduction lacks information on the used S. oralis strain. There are passages about other microbes, but the importance of this one is missing. The first paragraph of the results section could be inserted into the introduction.
- AUTHOR’S ANSWER: thank you very much for your comment; as suggested by the reviewers the first paragraph of the results section has been moved to the introduction. In this way we have also implemented the information about Streptococcus oralis. “Streptococcus oralis, a Gram-positive bacterium, is one of the early colonizers that interacting with the acquired pellicle deposited on biomaterials, provides the basis for the polymicrobial biofilm formation, with the subsequent colonization of facultative and obligate anaerobic microorganisms [23]. This shift in the composition of the microbial ecosystem could lead to a local host inflammatory response in the peri-implant tissues, and depending on the presence of other risk factors, it could lead to a reversible peri-mucositis or could also cause the peri-implantitis characterized by the irreversible bone loss [6,24,25]. Streptococcus oralis has been widely used for in vitro research on titaniumsurfaces. In this study, we present the effect of photodynamic therapy mediated by the use of a novel gel containing aminolevulinic acid and red LED irradiation (ALAD-PDI) on S. oralis biofilm, grown on MACHINED and DAE surfaces. This bacterium was chosen as a possible example of an oral microorganism and further studies with a more relevant mixture of bacteria would have to be done for any potential clinical applications”.
The conclusion section should be further developed and not only repeat the result. How it can be further used? Why it is important to apply the proposed solution? What are further stages if this idea goes to practice?
- AUTHOR’S ANSWER: thank you very much for your comment. The following sentence has been added to the text:
“If these results will be confirmed by further studies, this protocol could be proposed as a novel method for implant disinfection, of the fixtures affected by the peri-implant disease. Considering the antibiofilm activity shown by ALA-PDI, another possible application of this protocol could be the treatment of chronic wounds”.
I would like to point out also some minor issues L40 "gram-negative" should be in capital letter L65 "in vivo" and "in vitro" should be written in italics L233 "a test tube" - different font
- AUTHOR’S ANSWER: thank you very much for your comment. All suggested corrections have been made.